# The iNaturalist Sounds Dataset

**Mustafa Chasmai**[1]    **Alexander Shepard**[2]    **Subhransu Maji**[1]    **Grant Van Horn**[1]

[1]University of Massachusetts Amherst    [2] iNaturalist

{mchasmai, smaji, gvanhorn}@cs.umass.edu    alex@inaturalist.org

github.com/visipedia/inat_sounds

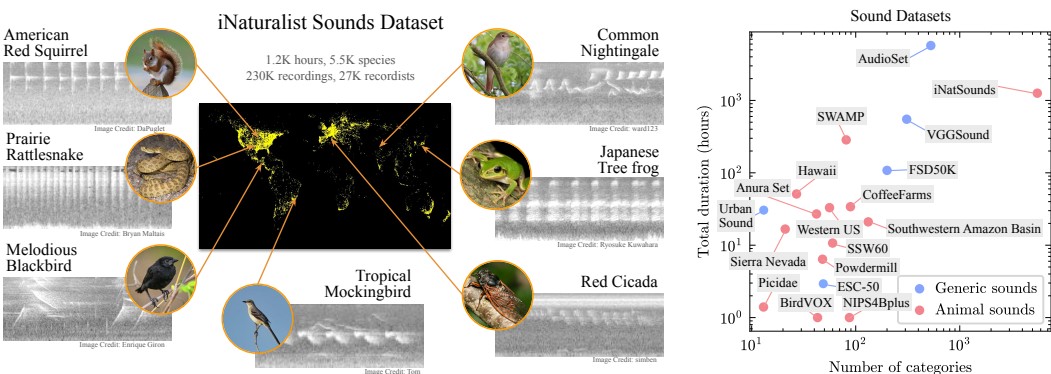

Figure 1: **Overview.** iNatSounds includes observations from around the world, capturing sounds of diverse species across major taxa. To the right, we show how iNatSounds compares to other general and animal specific sound datasets in terms of total audio duration and label diversity.

## Abstract

We present the iNaturalist Sounds Dataset (iNatSounds), a collection of 230,000 audio files capturing sounds from over 5,500 species, contributed by more than 27,000 recordists worldwide. The dataset encompasses sounds from birds, mammals, insects, reptiles, and amphibians, with audio and species labels derived from observations submitted to iNaturalist, a global citizen science platform. Each recording in the dataset varies in length and includes a single species annotation. We benchmark multiple backbone architectures, comparing multiclass classification objectives with multilabel objectives. Despite weak labeling, we demonstrate that iNatSounds serves as a useful pretraining resource by benchmarking it on strongly labeled downstream evaluation datasets. The dataset is available as a single, freely accessible archive, promoting accessibility and research in this important domain. We envision models trained on this data powering next-generation public engagement applications, and assisting biologists, ecologists, and land use managers in processing large audio collections, thereby contributing to the understanding of species compositions in diverse soundscapes.

## 1   Introduction

Fine-grained visual categorization has benefited from an abundance of benchmark datasets [4, 64, 71, 87, 90, 94]. These datasets enable systematic evaluation, allowing researchers to identify and improve specific bottlenecks, leading to significant progress. Research has focused on architectures for better transfer [6, 37, 55], techniques for handling rare classes associated with long-tail dataset

38th Conference on Neural Information Processing Systems (NeurIPS 2024) Track on Datasets and Benchmarks.

distributions [34, 52, 82], and human-in-the-loop recognition systems [5, 48], among others [49, 60, 97]. Continued progress has been due, in part, to further scaling up fine-grained datasets to keep pace with modern research trends. These recent datasets rely on high-quality community resources. For example, the iNat2021 [88] dataset, featuring 2.7 million images from 10,000 different species, was constructed from observations on the iNaturalist platform [36], a global community of citizen scientists. Models trained on the iNat2021 dataset generalize better to a variety of fine-grained recognition tasks in natural domains compared to the de facto standard ImageNet dataset [88]. Similar large-scale, fine-grained datasets for the audio domain are lacking in comparison.

Existing audio datasets enable researchers to explore a variety of tasks: sound event classification [24, 27, 75, 80], acoustic scene classification [38, 68], speech emotion and sentiment recognition [59, 77, 100, 101], music analysis [17, 23], audio captioning [21, 44, 65, 66], audio question answering [56], and audio retrieval [19, 47], among others. Large-scale audio datasets, such as AudioSet [27], have coarse category labels that are more akin to ImageNet [18] categories than to the species level categories found in fine-grained visual datasets like iNat2021. A current gap in the acoustic space is a large-scale, fine-grained dataset of species sounds that is easy to download and easy to use. AudioSet only provides URLs to YouTube videos (a portion of which are no longer available [24]), and while small-scale animal datasets exist (e.g., [30], see Fig. 1) , they lack the generalization benefits and research challenges that come from large-scale datasets.

Detecting, classifying, and studying animal sounds (bioacoustics) represents a crucial area of study for natural history and conservation purposes [76], and therefore should receive attention from the broader machine learning community. Automated methods for analyzing acoustic data can enable engaging public outreach applications [67], allow scientists to study animal compositions and behavior [3, 99], and assist land use managers and conservationists in identifying species within habitats to make informed decisions about remediation and restoration [45, 53, 86, 95].

To help fill the current gap in large-scale, fine-grained acoustic datasets and help make this research area more accessible, we present the iNaturalist Sounds Dataset (iNatSounds), see Fig. 1. The dataset is comprised of 230K audio files (∼1.2K hrs) capturing sounds from over 5,500 species and is sourced from iNaturalist [36]. We benchmark multiple backbone architectures on iNatSounds and compare multiclass and multilabel training objectives. With Top-1 accuracy in the low 60% range, there is ample room for improvement by the research community. We demonstrate the immediate utility of iNatSounds and our training protocols by benchmarking on downstream evaluation datasets. iNatSounds, including audio, annotations, and licensing info is available here.

## 2 Related Work

**Sound event classification datasets.** The acoustic research community has built multiple datasets for sound event classification [24, 27, 75, 80]. This challenge has been explored extensively in the DCASE workshops [16] with smaller scale datasets like UrbanSound8K [80] and ESC-50 [75] composed of urban and general sound categories. The AudioSet dataset [27] significantly increased the dataset size with 1.7M human-labeled 10s sound clips drawn from YouTube videos covering 527 event classes. The event classes were sampled from the AudioSet Ontology[27] and cover everyday sounds like "keys jangling" and "acoustic guitar," which are grouped into super-categories like "home sounds" and "musical instruments." A downside to this dataset is that the original content was not licensed for research use, and therefore only YouTube URLs are released (other datasets like VGGSound [9] suffer similar downsides). This has obvious pitfalls, primary of which is that YouTube videos can be deleted. This was pointed out by the authors of the FSD50K dataset [24], who used 200 categories from the AudioSet ontology to build an easy to use dataset of 50K appropriately licensed recordings sourced from Freesound [25]. Freesound has become a standard source of data for acoustic datasets [26] due to their licensing transparency. iNatSounds combines the large-scale nature of AudioSet, with the ease of data accessibility provided by datasets like FSD50K.

**Bioacoustic classification datasets.** The Macaulay Library [62] and XenoCanto [96] are two popular sources of species media (sounds, images and more, particularly focused on birds), similar to iNaturalist [36]. However, all three resources act as archives, providing the raw resources to build datasets, as opposed to being datasets in and of themselves. BEANS [30] presents a benchmark of 12 public datasets covering various species including birds, bats, and mosquitoes. While they standardize evaluation, the scale of the benchmark is relatively small, with a maximum of 264 species per dataset.

Other existing benchmarks for birds [12, 15, 32, 33, 42, 58, 69, 70, 81, 91, 93], bats [79, 103], and frogs [8] are also relatively small. SWAMP [41] consists of 285 hour-long soundscape recordings captured in Sapsucker Woods (SSW) in upstate New York, USA. All bird vocalizations, totaling 81 different species, have been annotated in these recordings. SSW60 [89] contains curated multimodal data (audio, images, and video) for each of its 60 bird species. Powdermill [11] covers 48 bird species recorded in the Powdermill Nature Reserve in Pennsylvania, USA. These datasets are focused on the Northeastern US and capture only a small portion of the entire spectrum of birds, let alone millions of other taxa, from around the world. Competition datasets from LifeCLEF [54] and DCASE [16] face similar constraints. Concurrent to our work, BirdSet [78] includes recordings from XenoCanto and a set of downstream tasks for evaluation. iNatSounds scales up the number of training species to over 5K geographically diverse taxa. We compare iNatSounds with other datasets in Fig. 1.

**Multilabel classification from single positive labels.**    As discussed in § 3, iNatSounds is weakly labeled. Building on prior research for learning from single-positive labels [13], we explore methods for training a multilabel classifier. These approaches employ various strategies to handle "unknown" labels, such as maximizing entropy [105] or assuming absence [7, 39, 50, 61, 63, 83]. We find the "assume absent" strategy effective for audio, although performance drops compared to a multiclass classifier when the evaluation dataset is actually single label. Nevertheless, we demonstrate that performance can be recouped by using semi-supervised learning techniques based on student-teacher models [84], resulting in models that perform well across both single and multilabel tasks.

**Audio recognition models.**    While classical audio recognition models relied on hand-crafted features [85], recent efforts have shifted focus to deep learning approaches [102]. Sequence models like RNNs [1, 10, 72, 98] and Transformers [74, 92] have been used for audio recognition, using raw audio signals as sequential inputs. A common strategy in prior art is to represent audio as a spectrogram over different frequencies, treat the 2D spectrogram as an image and apply visual recognition models [22, 40, 89]. Slow-Fast networks [43] use spectrograms at two time scales for better temporal context. Recent methods also explore the benefits of pretraining to learn few and zero-shot generalizable audio embeddings using supervision of frequent classes [28], paired language supervision [22], and self-supervision [29]. We benchmark three architectures common in vision research and show the effectiveness of pretraining on iNatSounds for strong downstream performance even without any finetuning. The very long tail of iNatSounds species distribution (Fig. 2) provides an opportunity for future research on few-shot learning in this domain.

## 3   iNatSounds Construction

**Initial filtering.**    iNatSounds is sourced from iNaturalist [36], a global citizen science platform of species observations consisting of media, location information, and time metadata. iNaturalist uses majority vote to identify (i.e., "label") observations to various taxonomic levels in the tree of life. Observations that reach a consensus identification at the rank of genus or lower, and have valid metadata fields, are given a "research-grade" badge. See [57] for an analysis of the quality of these identifications. We begin the constructions of iNatSounds by filtering an export of research grade observations, taken on February 1, 2024, to those that contain audio media and have been identified to species, resulting in 360K observations. We further filter these observations to those belonging to 5 taxonomic classes where bioacoustics is often studied: Aves, Insecta, Reptilia, Mammalia, and Amphibia. An observation can have multiple audio files associated with it, but we only keep the primary recording for iNatSounds. Audio files uploaded to iNaturalist come in a wide range of formats. For consistency and ease of use, we remove observations with audio having a sampling rate exceeding 48kHz, a common cutoff for microphones in mobile phones due to human perception limitations, and resample the remaining audio files to 22.05 kHz and store them as single-channel WAV files. This leads to a candidate set of 348K observations spanning 7,092 species.

**Train, validation and test splits.**    We construct the train, validation, and test splits using a simple process that is meaningful and easy to repeat with new data availability: we split the observations by year. Observations made during or before 2021 are eligible for the train split, and observations made in 2022 and 2023 are eligible for the validation and test splits respectively. Since most natural migrations follow yearly cycles (and species can sound different throughout these cycles), we expect test data spanning an entire year to be representative of a species acoustic repertoire. The dataset is easy to update as new observations become available: next year's data will become the new test split, current test split will become the new validation split and current validation split will be added to the

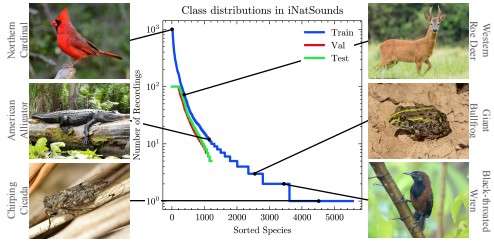

| Class | Train | | Val/Test | Val | Test |
| | Species | Audio | Species | Audio | Audio |
| --- | --- | --- | --- | --- | --- |
| Aves | 3,846 | 111,029 | 939 | 37,597 | 41,036 |
| Insecta | 745 | 10,080 | 111 | 3,065 | 3,305 |
| Amphibia | 650 | 13,183 | 118 | 4,004 | 4,081 |
| Mammalia | 296 | 2,566 | 41 | 983 | 1,073 |
| Reptilia | 32 | 154 | 3 | 49 | 32 |
| **Total** | 5,569 | 137,012 | 1,212 | 45,698 | 49,527 |

Figure 2: **Dataset Statistics.** Distribution of the number of recordings for each species in the train, validation, and test splits of iNatSounds. Some frequent and less frequent species are also shown. Note that the training set has a larger number of species than the validation and test splits.

train split. With this splitting strategy, 147K, 91K and 106K observations are eligible for the train, validation and test splits respectively.

**Selecting species.**    While assigning observations to a split based on year is easy to implement, it does have one big drawback: there is no guarantee that a species will be observed each year by the iNaturalist community. We therefore only include a species in validation and testing if it has at least 5 observations in all three splits. This results in 1212 species to be used for evaluation. We do not restrict the train split to these 1212 species, but rather include all 5569 species that have at least 1 observation in the train split. In our experiments, we train with all 5569 species and then ignore species outputs that do not overlap with the 1212 val/test species when doing evaluation. Increased species coverage in the train split allows for evaluation on more downstream datasets and provides more data to train foundation models that can be fine-tuned. This additional data can also be helpful in training predictors of higher taxonomic levels (i.e., by rolling observations up to the rank of genus or family, etc.).

**Further subsampling.**    The previous steps result in more than 80K sounds in both the validation and test splits. To reduce evaluation time, we subsample the validation and test splits using a spatio-temporal clustering approach to maintain diversity across a species range and seasonal variation. We first cluster observations of each species in each split according to geographic distance and time intervals between individual observations. We then randomly sample observations from the *clusters* in a round-robin fashion till we sample at most 100 observations for each species. For species with less than 100 observations to begin with, we keep all samples in the split. For the train split, we cap species at 1K observations using the same technique. This results in 137K, 45K and 49K observations in the train, validation and test splits respectively. The distribution of observations for the three splits, along with statistics for the 5 taxonomic classes can be found in Fig. 2. While our sampling strategy aims to construct training and evaluation sets representative of the temporal and geographic diversity of species, we acknowledge that we can not completely escape the inherent biases of data archived on iNaturalist. For example, Fig. 1 clearly shows a bias towards US and European locations.

**Evaluation metrics.**    We compute evaluation metrics at the "file-level." Regardless of a file's duration, a method is expected to output one score value for each of the 1212 evaluation species. We report class-averaged Top-1 and Top-5 accuracy, as well as class-averaged mAP and mF1, where we optimistically choose the best threshold per species when computing F1. mAP is the primary metric for iNatSounds.

## 4   iNatSounds Models

We take a vision approach to classifying sounds: we convert a 1D audio waveform into a 2D spectrogram through the Short-Time Fourier transform (STFT). This applies the Fourier transform to windowed sections of the 1D audio waveform, resulting in a 2D matrix (a "spectrogram") that contains frequency spectra (rows) over time (columns), with a trade off in resolution between the two axes controlled by the window size and stride. This transformation is bijective, one can reconstruct the original audio signal with the inverse STFT. We treat this 2D spectrogram as an image and use image-based neural networks to classify species sounds from visual patterns in the spectrogram.

**Model inputs: converting audio to an image.** We follow a similar recipe as prior work [9, 89]. Assuming single-channel audio with a sample rate of 22.05kHz, we use a Hann window size of 512, with a stride length of 128, and 1024 FFT length for the STFT. We convert the linear spaced frequencies to mel-scale to better align with human perception of pitch change. Our mel-scale maps frequencies in the range [50Hz, 11.025kHz] to 128 logarithmically spaced mel bins. The resulting magnitude values are converted to decibels. We convert to a traditional gray scale image by rescaling the decibel values to span [0, 255] and save the matrix as an uint8 image. This process converts one second of audio into an image that is approximately 168×128 (width×height); three seconds of audio produces an image of size ∼512×128. See Figs. 1, 3, and 4 for example spectrogram images.

Audio files can have variable duration, and therefore our spectrogram images can have variable width. Simply resizing all spectrograms to a fixed size would severely compress information in long recordings. Instead we stride the audio by a fixed sized window. The input to our models is fixed at ∼3 seconds of audio (512 pixels). To handle a long audio file, we process 3 second windows that are strided by 1.5 seconds (256 pixels). If an audio recording is less than the window size, we pad with silence (0s) to 3 seconds. To make use of ImageNet pretrained backbones, we duplicate the gray scale spectrogram image to create an RGB image. Transformer backbones, like ViT-B-16 [20], have a fixed input resolution of 224×224. To standardize the input across experiments, we use bilinear interpolation to resize the 512×128×3 RGB spectrograms to 224×224×3 for all models.

**Data augmentations.** Following prior work [73, 89], we do frequency masking (erase 15 random consecutive bins out of 128 frequency bins) and time masking (erase 50 random consecutive bins out of 512 time bins). We also use Mixup [104], taking a weighted average of windows from different recordings. Since the spectrograms are in decibels, we exponentiate before averaging and get the logarithm after. For multiclass training, we use a weighted average of the one-hot labels of the two windows, while for multilabel we treat both species as present. The weight for averaging is sampled from a beta distribution $B(\alpha = 1, \beta = 1)$.

**Multiclass and multilabel learning.** Our dataset $\mathcal{D} = \{(\mathbf{x}_i, \mathbf{y}_i)\}_{i=1}^N$ consists of pairs of audio $\mathbf{x}_i$ and label $\mathbf{y}_i = \{0, 1\}^C$ indicating the presence or absence of each of $C$ classes. However, iNatSounds is weakly labeled as it contains a single positive label for the entire audio file. Thus it is possible that a particular 3-second window $\mathbf{w}_i$ within the recording $\mathbf{x}_i$ might contain only background noise or, worse yet, another species that is part of the training set. Our models $f(\mathbf{w})$ operate on fixed sized windows of the inputs. The simplest approach is to treat it as a multiclass classification problem and train $f$ using the cross entropy loss. In particular we optimize $L_{CE}$ defined over a batch $\mathcal{B} = \{(\mathbf{x}_i, \mathbf{y}_i)\}_{i=1}^B$ of recordings and labels as

$$L_{CE} = \frac{1}{B} \sum_{i=1}^B \left( \sum_{c=1}^C \mathbf{y}_i^c \log \left[ \texttt{softmax} \left( f(\mathbf{w}_i) \right)^c \right] \right)_{\mathbf{w}_i \sim \mathbf{x}_i}, \tag{1}$$

where $\texttt{softmax}$ is defined across $C$ classes and $\mathbf{w}_i$ is a randomly sampled window within $\mathbf{x}_i$ with data augmentation. While this model predicts a single positive class within a window, multiple predictions can be obtained for the entire file by aggregating predictions over windows.

To learn a classifier which can natively handle multiple predictions within a window, i.e., a multilabel classifier, we incorporate two strategies. First, based on techniques for learning from single-positive labels [13], we use a simple approach that assumes all unlabelled species to be negatives. Concretely, the assume-negative loss $L_{AN}$ for training classifier $f$ is given by

$$L_{AN} = \frac{1}{B} \sum_{i=1}^B \left( \sum_{c=1}^C [\mathbf{y}_i^c] \log[\sigma \left( f(\mathbf{w}_i) \right)^c] + [1 - \mathbf{y}_i^c] \log[1 - \sigma \left( f(\mathbf{w}_i) \right)^c] \right)_{\mathbf{w}_i \sim \mathbf{x}_i}, \tag{2}$$

where the sigmoid function $\sigma$ replaces the earlier $\texttt{softmax}$ function. More variations of this loss function have been proposed in prior art [13], involving subsampling or reweighting negatives, but we found this scheme to be effective.

Second, taking inspiration from semi-supervised classification methods, we propose the use of a student-teacher [84] framework to pseudo-label the data. The teacher $f_T$ is initialised by a multilabel model trained with $L_{AN}$ to provide supervisory signal to the student. The student and the teacher receive different augmentations of each window. The student is trained with a combination of losses

from the ground truth labels and the pseudo-labels. The final loss is given by

$$L_{ST} = \frac{1}{B} \sum_{i=1}^{B} \left( L_{AN}(f(\mathbf{w}_i), \mathbf{y}_i) + \lambda \|\sigma(f(\mathbf{w}_i)) - \sigma(f_T(\mathbf{w}_i))\|^2 \right)_{\mathbf{w}_i \sim \mathbf{x}_i}, \tag{3}$$

where $\lambda$ is a weighting parameter. Parameters of the teacher $\theta_T$ are updated as exponentially moving averages of the student parameters $\theta$, given by $\theta_T' = \alpha\theta_T + (1 - \alpha)\theta$, where the decay parameter $\alpha \in [0, 1]$ controls the rate of teacher updates. We observe best performance with $\alpha = 0.999$ and $\lambda = 1000$. For the augmentations, after mixup we perform two different random erasures for the student and the teacher, with each erasure removing 10-33% area of a window. For test evaluation we take whichever model, student or teacher, performs best on validation.

**Model backbones and train/test configurations.** We experiment with MobileNet-V3-Large [35], ResNet-50 [31] and ViT-B-16 [20] for our classification models. This covers a wide range of model sizes from 6.24M of MobileNet to 87.02M of ViT. We initialise each model with ImageNet [18] pretrained weights. During training, we randomly sample one window from each train recording per epoch. This means recordings of different duration contribute equally to a training epoch. We train for 50 epochs and do early stopping, choosing the checkpoint with best validation performance. The learning rate (lr) is ramped up from 10% to 100% of the maximum lr over the first 5 epochs and then decayed back to 10% with a cosine schedule over the remaining 45 epochs. For MobileNet and ResNet50, we use SGD with Nesterov acceleration and a maximum lr of 0.05, while for ViT, we used the Adam optimizer with a maximum lr of $10^{-4}$. During evaluation, we produce file-level predictions by averaging predicted probabilities on all strided windows (3s window, 1.5s stride) of a recording.

**Incorporating geographic priors.** Prior work used the location information associated with iNaturalist observations to infer geographic ranges of species around the world [14, 51]. We can benefit from these techniques during evaluation by filtering out species that are unlikely to occur at the location of a given observation [61]. We use the SINR framework [14] to train a range estimation model for the 5,569 species in iNatSounds using observation data from the same iNaturalist database export used to build iNatSounds. However, instead of using observations with audio, we use the significantly larger set of observations with images. SINR predicts the probability of observing a species given a location. For each recording, we use the associated location to generate a binary mask by thresholding SINR probability of each species (a threshold of 0.1 performs best on the validation set). These masks are used to filter iNatSounds model predictions.

## 5 Experiments

### 5.1 Multiclass Performance on iNatSounds

Table 1 shows Top-1 and Top-5 accuracy for our three backbone variants trained using the multiclass objective, i.e., $L_{CE}$ in Eq. 1. We observe the expected trend of increasing model complexity leading to increased Top-1 test performance: 49.1→52.6→53.6 for MobileNet-V3, ResNet-50, and ViT-B-16 respectively. Utilizing geopriors to filter out geographically irrelevant species boosts Top-1 accuracy for all methods, with ResNet-50 achieving the best performance: 56.9→60.7→60.3. It is interesting to note that MobileNet-V3 combined with SINR (two very small models) outperforms the much larger ViT-B-16 model, 56.9 vs 53.6.

Fig. 3 provides additional insights into our experiments. In Fig. 3 (right) we show Top-1 validation performance for species binned by training sample size. We observe a clear trend of increasing performance as the number of training samples increases: mean Top-1 performance of ∼19 vs ∼80 for species with 5-10 train samples vs those with 500-1K samples. Even with large train sets (200-500 samples), there are still some outliers with low performance (∼40 top-1 accuracy), hinting at fine-grained acoustic challenges. In Fig. 3 (left) we show four of the most confused species pairs along with one of their images and sounds. Note the similarity of the images and the spectrograms.

### 5.2 Transfer to Downstream Bioacoustic Datasets

In this section we demonstrate the utility of iNatSounds as a bioacoustic foundation dataset and benchmark its performance on several downstream datasets.

Table 1: **iNatSounds Results.** Class averaged Top-1 and Top-5 accuracy on the val and test splits for multiclass models with different backbones. Geo-prior attempts to filter out geographically irrelevant species, see Sec. 4. Each experiment is repeated with three different seeds and mean ± std is reported.

| Model | Parameters | Geo Prior | Validation Set | | Test Set | |
|-------|-----------|-----------|-------|-------|-------|-------|
| | | | Top-1 | Top-5 | Top-1 | Top-5 |
| MobileNet-V3 [35] | 6.24M | ✗ | 50.2 ± 0.23 | 70.5 ± 0.06 | 49.1 ± 0.17 | 69.5 ± 0.18 |
| ResNet-50 [31] | 26.77M | ✗ | 53.4 ± 0.96 | 74.6 ± 0.41 | 52.6 ± 0.95 | 74.3 ± 0.59 |
| ViT-B-16 [20] | 87.02M | ✗ | 55.1 ± 0.04 | 74.9 ± 0.40 | 53.6 ± 0.22 | 74.3 ± 0.32 |
| MobileNet-V3 [35] | 6.24M | ✓ | 58.1 ± 0.22 | 78.1 ± 0.22 | 56.9 ± 0.19 | 77.6 ± 0.26 |
| ResNet-50 [31] | 26.77M | ✓ | 61.3 ± 0.56 | 81.3 ± 0.25 | 60.7 ± 0.62 | 81.3 ± 0.63 |
| ViT-B-16 [20] | 87.02M | ✓ | 62.0 ± 0.23 | 80.9 ± 0.63 | 60.3 ± 0.27 | 80.4 ± 0.43 |

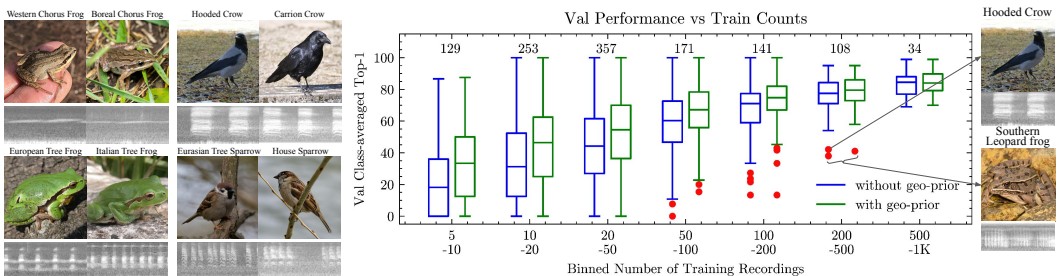

Figure 3: **Analysis of iNatSounds Performance.** Left: Sampled top confusion pairs highlighting fine-grained acoustic challenges. Right: Class-averaged Top-1 accuracy on iNatSounds val set, binned by training set size. We show plots with and without geo-priors (used to filter out geographically irrelevant species, see Sec. 4) and explore outliers for classes with a high number of training recordings (200-500). Both figures are from a multiclass MobileNet model.

**Datasets and evaluation details.** We benchmark on three avian datasets: SSW60 [89], Powdermill [11], SWAMP [41]; and one frog dataset AnuraSet [8]. SSW60 is weakly labeled, and therefore we follow the same evaluation process for iNatSounds: we average the predictions of all 3-second windows (strided by 1.5-seconds) from a given test sample to arrive at the final, "file-level" scores for each species. SSW60 is relatively class balanced and reports Top-1 accuracy across samples instead of class averaged Top-1 accuracy. The remaining datasets are strongly labeled, meaning they have onset-offset annotations for each species sound event in a recording; see Fig. 4 for an example of these "boxes" rendered on the spectrograms. For these datasets we can determine the labels (i.e., species) present in each 3-second window of audio and therefore can treat them as a multilabel prediction problem. A window is labeled with all species that have a "box" that overlaps the window by at least 10% (or vice-versa: the window overlaps a box by at least 10%). To evaluate on these datasets we use average precision (AP) and max F1 scores (F1) computed across every 3-second window of audio, strided by 1-second. We average these metrics across all species in the dataset to arrive at mean average precision (mAP) and mean F1 (mF1). iNatSounds contains all bird species annotated in the three bird datasets, and we evaluate the performance of pre-trained iNatSounds models by simply masking out species scores not present in each downstream dataset. Of the 42 frog species present in AnuraSet, 25 are present in the training set of iNatSounds and we evaluate on the subset of AnuraSet having only these species (again using pre-trained iNatSounds models). No fine-tuning is done on these downstream datasets.

**Multiclass vs Multilabel classification.** In Table 2 we compare our multiclass and multilabel objectives using different backbones on iNatSounds and our downstream datasets. While a multiclass objective maximizes top-1 and top-5 accuracy on iNatSounds, we observe that models trained with a multilabel objective transfer better to downstream datasets. We can further improve the performance of our convolution backbones (MobileNet-V3 and ResNet50) by using a mean teacher [84] training strategy. However, this strategy does not achieve the best results on downstream datasets when using a transformer backbone. In Fig. 4 we visualize the outputs of models trained using a multiclass and a multilabel strategy across the various datasets. These visualizations provide intuition for why

Table 2: **Multiclass and Multilabel experiments.** Each experiment repeated thrice and mean$^{\text{std}}$ reported for iNatSounds and downstream. MV3: MobileNet-V3, R50: ResNet50. ML: Multi-Label and MT: Mean Teacher . * test set includes more species. † likely trained on test files.

| Model | ML | MT | iNatSounds test | | | | SSW60 | | Powdermill | | SWAMP | | AnuraSet | |
|---|---|---|---|---|---|---|---|---|---|---|---|---|---|---|
| | | | mAP | mF1 | Top1 | Top5 | Top1 | Top5 | mAP | mF1 | mAP | mF1 | mAP | mF1 |
| Reported SOTA | | | - | - | - | | 67.4 | - | - | - | - | - | - | **37.8** * |
| BirdNET | | | - | - | - | | **86.7**$^\dagger$ | **94.6**$^\dagger$ | **60.0**$^\dagger$ | **60.7**$^\dagger$ | **37.5**$^\dagger$ | **41.7**$^\dagger$ | - | - |
| Merlin Sound ID | | | - | - | - | - | - | - | **62.9** | **64.8** | **50.4** | **54.2** | - | - |
| MV3 | ✗ | ✗ | $60.0^{0.3}$ | $63.2^{0.3}$ | **$49.1^{0.2}$** | **$69.5^{0.2}$** | $66.9^{0.9}$ | $83.9^{1.9}$ | $35.0^{2.2}$ | $38.6^{2.9}$ | $32.9^{1}$ | $37.9^{0.9}$ | $29.9^{0.2}$ | $33.9^{0.8}$ |
| | ✓ | ✗ | $57.3^{0.6}$ | $60.7^{0.6}$ | $42.4^{0.4}$ | $63.5^{0.2}$ | $61.4^{1.3}$ | $80.0^{1.1}$ | $36.4^{0.7}$ | $39.8^{0.4}$ | $31.3^{0.3}$ | $35.7^{0.2}$ | $31.3^{1.3}$ | $36.2^{0.8}$ |
| | ✓ | ✓ | **$61.0^{0.3}$** | **$63.9^{0.2}$** | $46.5^{0.3}$ | $67.3^{0.5}$ | **$68.4^{1.2}$** | **$84.0^{0.1}$** | **$37.1^{2.1}$** | **$40.2^{1.9}$** | **$33.9^{1.6}$** | **$38.6^{1.4}$** | **$31.8^{0.4}$** | **$36.4^{0.2}$** |
| R50 | ✗ | ✗ | $62.8^{0.7}$ | $65.5^{0.4}$ | **$52.6^{1.0}$** | **$74.3^{0.6}$** | $68.1^{1.7}$ | $83.2^{1.2}$ | $38.0^{0.9}$ | $41.3^{0.7}$ | **$35.9^{0.8}$** | **$41.7^{0.8}$** | $28.9^{1.2}$ | $33.4^{0.4}$ |
| | ✓ | ✗ | $60.5^{0.8}$ | $63.5^{0.6}$ | $48.1^{0.4}$ | $69.1^{0.5}$ | $63.8^{1.4}$ | $82.2^{0.7}$ | **$39.7^{1.2}$** | **$42.9^{1.5}$** | $31.6^{1.2}$ | $37.0^{1.2}$ | $29.2^{1.2}$ | **$34.1^{1.8}$** |
| | ✓ | ✓ | **$63.5^{0.8}$** | **$66.3^{0.6}$** | $52.2^{0.4}$ | $72.0^{0.3}$ | **$70.4^{0.9}$** | **$85.7^{0.3}$** | $38.4^{0.9}$ | $42.1^{0.6}$ | $33.3^{0.7}$ | $38.9^{0.6}$ | **$29.7^{0.1}$** | $33.9^{0.3}$ |
| ViT | ✗ | ✗ | $64.4^{0.3}$ | $66.9^{0.2}$ | **$53.6^{0.2}$** | **$74.3^{0.3}$** | $70.9^{0.4}$ | **$85.8^{0.3}$** | $33.8^{0.8}$ | $37.8^{0.6}$ | $32.6^{1.1}$ | $38.5^{1.1}$ | $31.2^{2.3}$ | $35.2^{2.3}$ |
| | ✓ | ✗ | $65.0^{0.3}$ | $67.4^{0.2}$ | $52.4^{0.1}$ | $73.2^{0.2}$ | $70.3^{0.7}$ | $84.4^{0.5}$ | **$40.2^{0.9}$** | **$43.5^{1}$** | **$39.0^{0.9}$** | **$43.8^{0.9}$** | **$33.8^{2.0}$** | **$37.0^{2.0}$** |
| | ✓ | ✓ | **$65.4^{0.1}$** | **$67.9^{0.1}$** | $53.1^{0.2}$ | $72.0^{0.4}$ | **$71.7^{0.1}$** | $84.4^{0.3}$ | $37.8^{0.5}$ | $41.5^{0.5}$ | $37.4^{0.8}$ | $43.2^{0.7}$ | $32.5^{0.7}$ | $36.1^{0.8}$ |

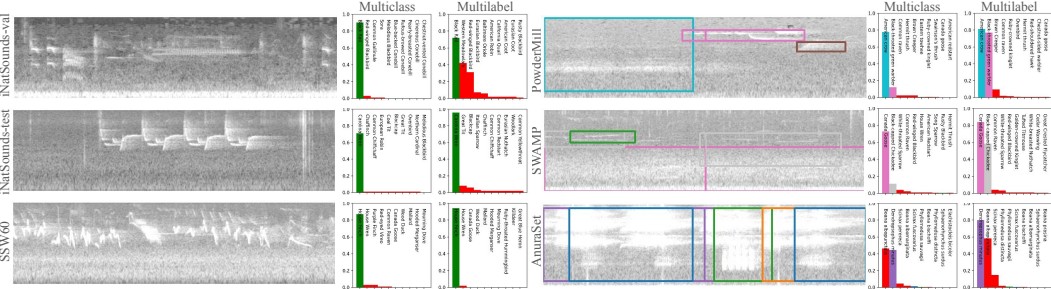

Figure 4: **Multiclass vs Multilabel Score Visualisations.** We compare the top 10 outputs from multiclass and multilabel MobileNet classifiers given 3-second inputs from weakly labeled (left) and strongly labeled (right) datasets. Box and bar colors are shared for the the strongly labeled datasets.

a multilabel objective does better at predicting multiple species (each with high confidence) on the strongly labeled downstream datasets, while a multiclass objective leads to single confident or multiple less confident predictions.

**Evaluation with weak labels.** The evaluation split of iNatSounds is weakly labeled, and therefore provides an imperfect picture of performance. Comparing a model's performance on iNatSounds to its performance on strongly labeled datasets like Powdermill, SWAMP, and AnuraSet provides an opportunity to determine how indicative performance on iNatSounds is for measuring progress on bioacoustic classification. The results in Table 2 suggest that exclusively using iNatSounds to measure the performance of a model is not recommended. While our MobileNet-V3 backbone shows a desirable correlation between iNatSounds performance and downstream performance, this pattern does not hold for our ResNet50 and ViT backbones; the best performing ResNet50 and ViT models on iNatSounds are not the best performing models on the downstream datasets. Therefore, we encourage researchers to report results on both iNatSounds and strongly labeled datasets when sharing models that could be used for biodiversity analysis. Because iNatSounds includes a large number of species, models trained on this dataset can often be directly applied to downstream datasets, simplifying the analysis. We present additional experiments with the BEANS [30] benchmark and a modified version of the BirdCLEF 2024 [46] challenge in the Appendix.

**Comparison to BirdNET.** We compare to the open source BirdNET model [40] (v2.4) on the downstream avian datasets; results shown in Table 2. BirdNET covers 6,000 species, including those found in these datasets, and also processes 3 seconds of audio, making comparison straightforward. The training dataset for BirdNET is not publicly documented nor is it made available. Correspondence

| Modality | iNatSounds training | SSW60 Videos [89] | Full | Reliable |
|---|---|---|---|---|
| Video-only | - | 76.2 | 75.4 | 77.6 |
| Audio-only | ✗ | 28.3 | 30.4 | 54.5 |
| | ✓ | - | 42.2 | 75.6 |
| Video+Audio | ✗ | 80.6 | 79.3 | 85.6 |
| | ✓ | - | **80.9** | **88.7** |

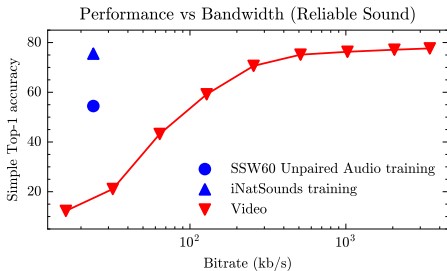

Figure 5: **SSW60 Multimodal Fusion.** Left: Top-1 accuracy (not class averaged) for different modalities from the SSW60 test dataset. Right: Top-1 accuracy (not class averaged) vs modality bitrate for video-only models and audio-only models. Both: DeIT for video, ViT multiclass for audio.

with the authors of BirdNET revealed that some portion of our downstream datasets are used to train their model. Therefore the results in Table 2 should be taken with a grain of salt. Regardless, we observe BirdNET achieving higher performance on SSW60 and Powdermill, and comparable performance on SWAMP. A key contribution of our work is the creation of a large-scale bioacoustics dataset, accompanied by a clear benchmarking procedure, promoting consistent and meaningful comparisons across studies.

**Comparison to MSID.**    The proprietary Merlin Sound ID (MSID) model [67] is trained on strongly labeled recordings from the Macaulay Library and covers ∼1.4k species. We've included MSID performance in Table 2 to serve as a reference point in contrast with our iNatSounds models, which encompasses over 5K species and are trained with weak supervision. While MSID's high performance demonstrates the power of strongly supervised training, it also highlights the considerable human effort required to achieve these results. The challenge to the research community, then, is clear: how can we approach or match high-performance, strongly supervised methods with minimal human labeling effort? Tackling this problem will be essential for creating large-scale, efficient bioacoustic models that are both accessible and impactful.

**Multimodal classification.**    SSW60 contains unpaired images and audio as well as videos with both modalities. We analyse performance of iNatSounds audio models fused with video-based models on SSW60 videos in Fig. 5 (left). Concretely, we extract image frames and audio separately from recorded videos, get predictions of each modality with corresponding models and then fuse (average) predicted scores to evaluate Video+Audio performance. The video model has a DeIT backbone and is pretrained on a subset of the iNat21 [88] image dataset before finetuning on video frames of the SSW60 train set. We fuse this video model with a ViT multiclass audio model either pretrained on iNatSounds or pretrained on ImageNet and finetuned on SSW60 unpaired audio, as described by [89]. Not all SSW60 videos have the target species producing sound, and we report performance on the subset of videos with species-identifiable audio as "Reliable" SSW60 videos. We achieve state-of-the-art audio-only and multimodal performance on SSW60.

The video component of SSW60 enables an interesting experiment not explored by the original authors [89]: how is Top-1 accuracy for different modalities affected by bitrate constraints? Scientists and practitioners placing sensors in remote locations have to account for bandwidth limitations; how should they prioritize modalities under different constraints? We explore this question in Fig. 5 (right) with video compression. Audio data allows classification with orders of magnitude lower bandwidth constraints and video performance drops sharply as we compress it to audio-comparable levels.

# 6   Conclusions, Limitations, and Future Work

We present iNatSounds, a large dataset of animal sounds covering 5,500 species across diverse geographic locations and taxonomic groups. We benchmark multiple backbone architectures and compare multiclass and multilabel classifiers. We observe multilabel objectives generally transfer better to strongly labeled downstream datasets even though a multiclass objective maximizes accuracy on iNatSounds. Our experiments also highlight the importance of evaluating models on strongly labeled datasets in addition to iNatSounds. While our MobileNet-V3 backbone shows desirable correlation between iNatSounds and downstream performance, this pattern does not consistently hold

across all architectures. Thus, we recommend researchers report results on both iNatSounds and strongly labeled datasets to ensure reliable assessments for biodiversity monitoring applications. The public availability of iNatSounds, combined with our benchmarking framework, invites the research community to advance fine-grained sound recognition and scalable bioacoustic analysis, particularly for models trained with weak supervision. Such progress will be critical for building impactful, large-scale bioacoustic models that maintain high performance with less human labeling effort.

The availability of weak labels is a limitation of the current dataset. While we are still able to train robust classifiers, stronger annotations with precise time-stamps of each sound would improve training and evaluation. Tools like Whombat [2] could help accelerate this process. iNatSounds also contains several biases; iNaturalist observations tend to be in geographically accessible regions and focused on North America and Europe. So species in other regions are not well represented. While the benefits of acoustic analysis for biodiversity monitoring are clear, there are privacy concerns as well as risks of these technologies being misused for poaching.

Observations in iNatSounds are licensed for research use, so we have not obfuscated or modified the data, but we should be respectful of this valuable resource. Personally identifiable information might be present or predicted from the observations. Location and time metadata is an obvious source of concern, and audio recordings might contain personally identifiable or inappropriate sounds.

Future work could involve a detailed evaluation of model architectures, model distillation, investigating pre-training strategies, incorporating geographic information during training, better multi-modal integration of audio and video, and continued scaling of the dataset.

## Acknowledgments and Disclosure of Funding

We wish to thank the iNaturalist community for sharing their data. The research is also supported in part by grants #2329927 and #1749833 from the National Science Foundation (USA).

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

# A   Appendix

Dataset documentation, licensing information, and metadata for iNatSounds are included along with the download links for audio recordings and annotations at `https://github.com/visipedia/inat_sounds`. For additional details on the quality control measures taken by iNaturalist, see `https://www.inaturalist.org/pages/archived+help#quality`. In what follows, we present some additional dataset statistics (Fig. A1), visualisations (Fig. A4 A3 A6 A5) and results (Table A1 A3).

## A.1   Additional Dataset Statistics

In Fig. A1, we start with a pie chart visualising the distribution of the number of species belonging to each iconic group (Fig. 2 of main paper). We also show a histogram of the durations of sounds in the dataset. Next in Fig A2, we show locations of all observations of some species in the dataset, highlighting the geographic diversity and coverage in iNatSounds. Finally, we show a visualisation of the sub-sampling method described in Sec. 3 of the main paper. All observations of a species in a split (blue) are clustered (orange cluster centers) according to geographic distance and time intervals between individual observations and then observations are randomly sampled from the clusters in a round-robin fashion (green).

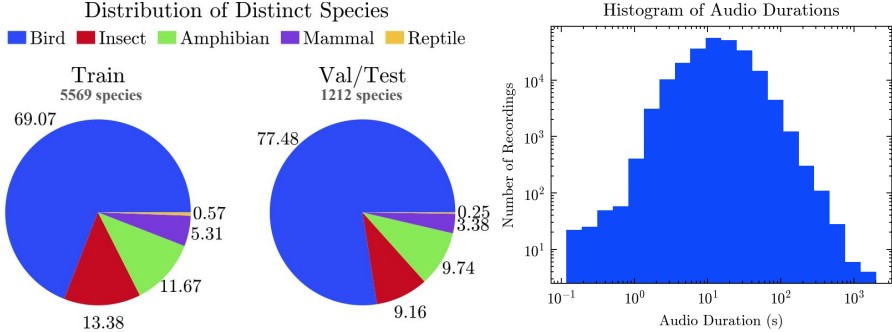

Figure A1: **Additional Dataset Statistics**. Left: Distribution of *distinct* species in the dataset according to different iconic groups (class level in taxonomy). Right: A histogram of audio durations for the entire dataset. Additional statistics:- mean: 19.52s, median: 13.94s, maximum: 33m 34.82s.

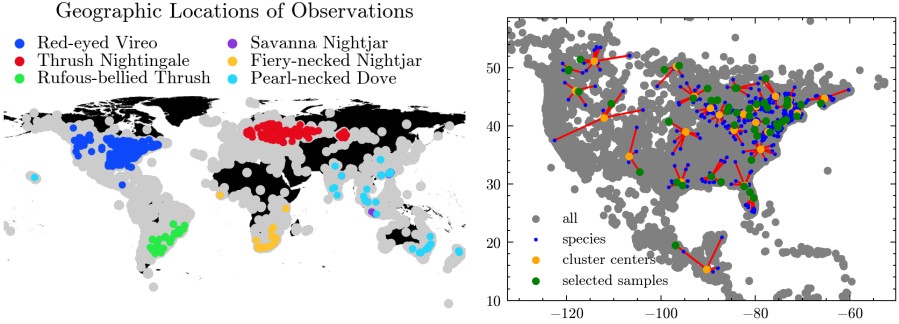

Figure A2: **Geographic Diversity and Sampling**. Left: Locations of observations in the training split. We show all observations of some species from around the world by colored points. Gray points are for the remaining species in the training split. Right: Sub-sampling data with clustering on the basis of geographic location and time of an observation.

## A.2   Confusions at different Taxonomy Levels

In Fig. A3, we show confusion matrices at different levels of taxonomy. We sort the classes according to their counts in the training set and group classes according to the iconic group. For the species level, we bin together 100 consecutive classes for the plot. Iconic groups are color-coded for easier

readability. One particular order of birds stands out with a large number of false positives. This is the order Passeriformes (Perching Birds), which is the largest order of birds.

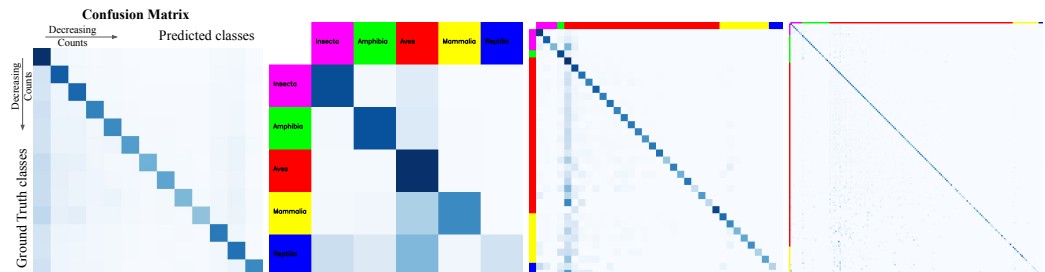

Figure A3: **Confusion**. Left: The normalised confusion matrix between predicted and ground truth classes. Classes are arranged in a decreasing order of their total number of train samples. For better clarity, we bin together 100 consecutive classes in each cell of the matrix. Next we have additional (normalised) confusion matrices for taxonomic class, order and family (left to right). For each of these, we use color maps insect: purple, amphibian: green, bird: red, mammal: yellow, reptile: blue.

## A.3 Analysis of the Geo Prior

In Fig. A4, we visualise the predicted range maps for the species shown in Fig. 1 of the main paper. The predicted probabilities (background color-map) and actual observations (gray circles) as well as the rough geographic locations (Fig. 1 main paper) are aligned well for all of these species.

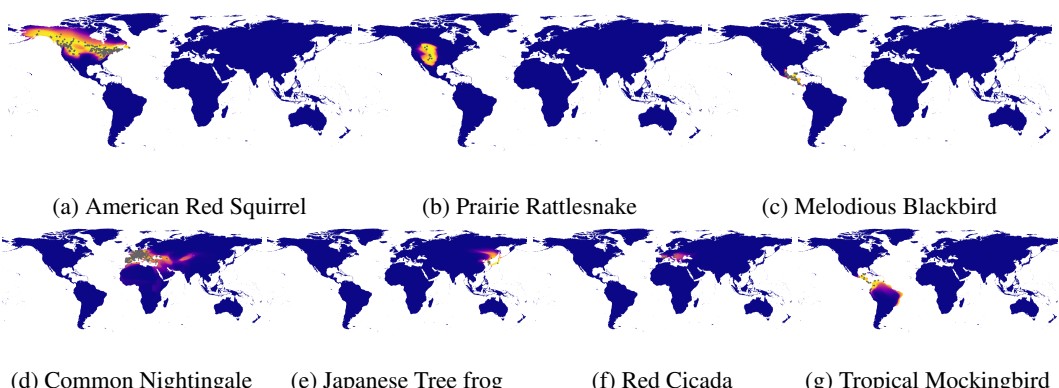

(a) American Red Squirrel  (b) Prairie Rattlesnake  (c) Melodious Blackbird

(d) Common Nightingale  (e) Japanese Tree frog  (f) Red Cicada  (g) Tropical Mockingbird

Figure A4: **SINR model range maps.** Predicted species ranges (probabilities) and actual geographic locations of recordings in iNatSounds (gray circles) for the species used in Fig. 1 (main paper).

Next, we further discuss the effects of incorporating geographic priors in iNatSounds models. We show the improvements for different iconic groups in Fig. A5. Reptiles show a big boost in performance, while the other groups have relatively uniform improvements. We list the species with most improvement by geo-priors in Table A1. We also show the predicted range maps and recorded geographic locations of these species in Fig. A6. For each pair, the range maps have significant differences, which is likely why the geo-prior helps these species. The very low performance without geo-priors (Table A1) hints at the acoustic similarity of these species and the improvements show the benefit of using such priors in fine-grained settings.

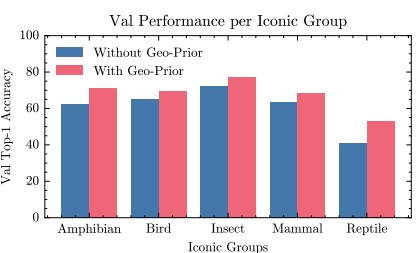

Figure A5: **Additional Analyses**. Per-iconic class-average accuracy on iNatSounds val set without (blue) and with (red) geo-priors.

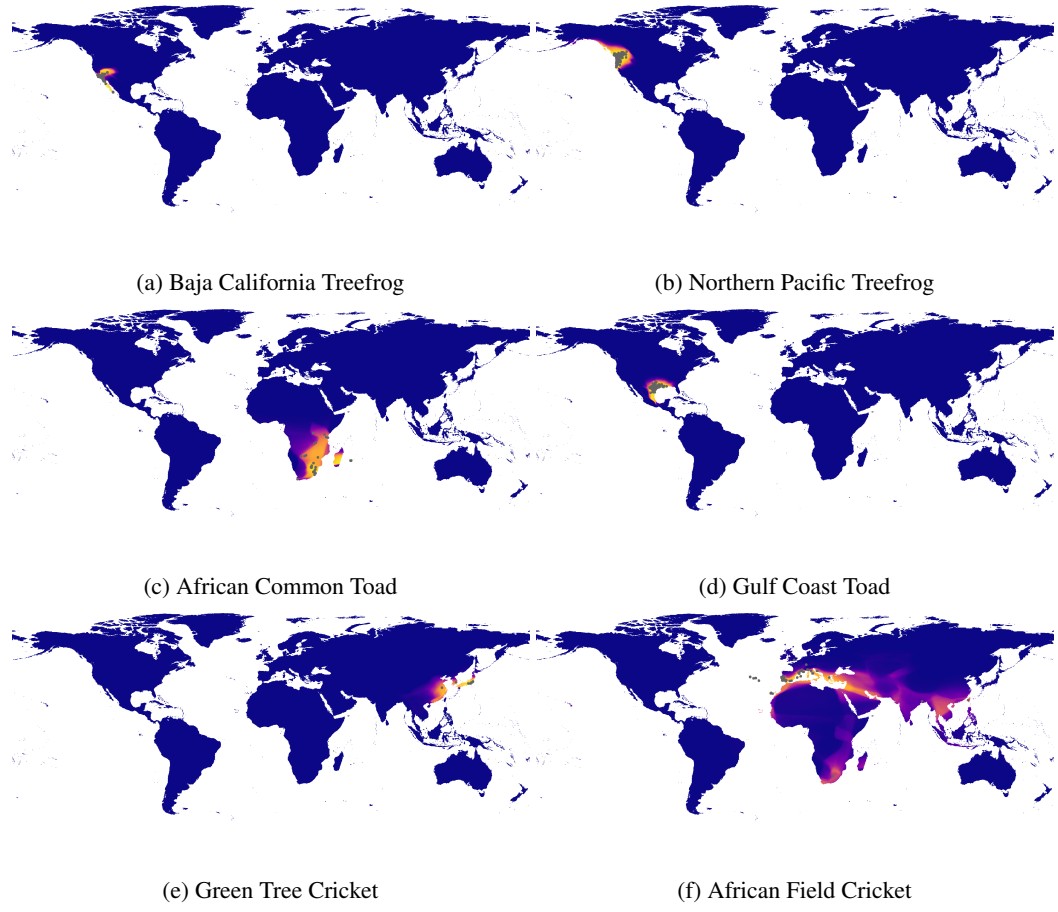

| (a) Baja California Treefrog | (b) Northern Pacific Treefrog |
| (c) African Common Toad | (d) Gulf Coast Toad |
| (e) Green Tree Cricket | (f) African Field Cricket |

Figure A6: **SINR model range maps.** Predicted species ranges (probabilities) and actual geographic locations of recordings in iNatSounds (gray circles). These are the species included in Table A1.

Table A1: **Classes most affected by geo-priors.**

| Class | without geo prior | with geo prior | Classes with the highest reduction in confusion |
|---|---|---|---|
| Baja California Treefrog | 22.5 | 85.0 | Northern Pacific Treefrog, Sierran Treefrog, American Bullfrog, Gray Treefrog, Savannah Sparrow |
| African Common Toad | 0.0 | 60.0 | Gulf Coast Toad, European Common Frog, Barred Owl, Ferruginous Pygmy-Owl |
| Green Tree Cricket | 0.0 | 57.1 | African Field Cricket, Carolina Ground Cricket, Tawny Owl, Fall Field Cricket |

## A.4 Masking Non-Eval Species

The train set contains 5569 species, while the evaluation set (both val and test) contain 1212 species. In Table A2, we explore the effects of masking train species that are not included in the evaluation split. As can be seen in the table, masking these species during evaluation boosts performance by a significant margin. Masking these species simulates the closed-set assumption used in most benchmarks, where the set of target classes are known ahead of time. For certain applications where the set of target species is not known, predicting without a mask may be a more realistic setting. In these applications, we often have access to the recording location, which would allow us to utilize a geo prior to automatically mask out species that are unlikely to occur at the given location. The last two columns of Table A2 show that using a geo prior actually achieves the best results.

Table A2: **Non-Eval Species Masking.** Here, we report iNatSounds performance with and without masking the $5569 - 1212 = 4357$ species that don't occur in the evaluation split. While this synthetic masking is potentially unrealistic in downstream tasks, knowledge of a recording's location is often available. We therefore include results that don't mask but rather filter based on a Geo-Prior. Using a Geo-Prior results in the best performance.

| Model | With Masking | | Without Masking | | No Mask, Geo - Prior | |
|---|---|---|---|---|---|---|
| | Top 1 | Top 5 | Top 1 | Top 5 | Top 1 | Top 5 |
| MobileNet | 50.2 | 70.5 | 43.8 | 63.7 | 53.2 | 73.2 |
| RresNet50 | 53.4 | 74.6 | 44.7 | 65.9 | 55.6 | 75.1 |
| ViT | 55.1 | 74.9 | 53.5 | 74.1 | 61.6 | 80.5 |

## A.5 ROC-AUC Metrics

In Table A7, we report ROC-AUC metrics corresponding to the experiments in Table 1 (main paper). ROC-AUC numbers are 99% for most models and it is hard to discern differences between the models by looking at this metric. In highly class imbalanced settings like ours, both the true positive rate (recall) and false positive rates tend to be quite low, which can lead to very high ROC-AUC scores. Precision recall curves are often more informative here, which is why we use mAP in Table 2.

| Model | Geo Prior | Val | Test |
|---|---|---|---|
| MobileNet | ✗ | 98.7 | 98.6 |
| ResNet50 | ✗ | 98.9 | 98.8 |
| ViT | ✗ | 99.0 | 98.9 |
| MobileNet | ✓ | 99.2 | 99.2 |
| ResNet50 | ✓ | 99.3 | 99.2 |
| ViT | ✓ | 99.4 | 99.4 |

Figure A7: **ROC-AUC.** We report ROC-AUC values for the experiments in Table 1 (main paper).

## A.6 Ablation on Mixup Augmentation

Finally, we ablate on the use of mixup as an augmentation. As can be seen in Table A3, mixup leads to significant improvements across all considered datasets.

Table A3: **Mixup Ablation.** Each experiment repeated thrice and mean$^{\text{std}}$ reported for iNatSounds and downstream. Models trained with multiclass objective. MV3: MobileNet-V3, R50: ResNet50

| Model | Mixup | iNatSounds test | | | | SSW60 | | Powdermill | | SWAMP | | AnuraSet | |
|---|---|---|---|---|---|---|---|---|---|---|---|---|---|
| | | mAP | mF1 | Top1 | Top5 | Top1 | Top5 | mAP | mF1 | mAP | mF1 | mAP | mF1 |
| MV3 | ✗ | $51.6^{0.2}$ | $56.7^{0.2}$ | $44.7^{1.3}$ | $65.9^{0.6}$ | $64.9^{1.9}$ | $83.4^{1.2}$ | $22.3^{0.8}$ | $25.7^{1.2}$ | $21.1^{1.4}$ | $26.3^{1.5}$ | $24.0^{0.7}$ | $29.2^{0.5}$ |
| | ✓ | $60.0^{0.3}$ | $63.2^{0.3}$ | $49.1^{0.2}$ | $69.5^{0.2}$ | $66.9^{0.9}$ | $83.9^{1.9}$ | $33.5^{1.1}$ | $37.7^{1.3}$ | $32.1^{0.7}$ | $38.1^{0.6}$ | $29.9^{0.2}$ | $33.9^{0.8}$ |
| R50 | ✗ | $55.5^{0.4}$ | $59.9^{0.4}$ | $48.1^{1.0}$ | $69.6^{0.3}$ | $67.2^{2.8}$ | $85.0^{2.0}$ | $24.1^{0.3}$ | $27.7^{0.6}$ | $25.0^{2.4}$ | $31.0^{2.6}$ | $24.4^{2.1}$ | $29.3^{1.7}$ |
| | ✓ | $62.8^{0.7}$ | $65.5^{0.4}$ | $52.6^{1.0}$ | $74.3^{0.6}$ | $68.1^{1.7}$ | $83.2^{1.2}$ | $36.5^{1.0}$ | $40.5^{0.6}$ | $33.9^{0.7}$ | $40.6^{0.8}$ | $28.9^{1.2}$ | $33.4^{0.4}$ |
| ViT | ✗ | $56.1^{1.5}$ | $60.2^{1.1}$ | $47.1^{1.0}$ | $69.2^{1.0}$ | $67.2^{0.1}$ | $85.2^{0.1}$ | $24.6^{1.3}$ | $28.2^{1.4}$ | $24.3^{2.2}$ | $30.4^{2.0}$ | $31.9^{3.2}$ | $35.3^{3.4}$ |
| | ✓ | $64.4^{0.3}$ | $66.9^{0.2}$ | $53.6^{0.2}$ | $74.3^{0.3}$ | $70.9^{0.4}$ | $85.8^{0.3}$ | $31.7^{0.2}$ | $35.7^{0.6}$ | $31.5^{1.1}$ | $37.9^{1.3}$ | $31.2^{2.3}$ | $35.2^{2.3}$ |

## A.7 Additional Downstream Datasets

We present additional experiments with the BEANS [30] benchmark and BirdCLEF 2024 [46] in Table A4 and Table A5 respectively. For the BEANS benchmark, some tasks were not species classification, but were related to it. Thus, iNatSounds models may not be directly applicable and we instead fine-tune these models on the respective train splits. For BirdCLEF, we construct random train-val splits from the released training data and compare pretraining strategies. Note the actual BirdCLEF 2024 competition has a hidden test set that we do not have access to.

Table A4: **Performance on BEANS [30]**. Performance on 5 datasets part of BEANS. These are models pre-trained with iNatSounds and fine-tuned on corresponding dataset train-set.

| Model | CBI Top1 | HumBugDB Top1 | Dogs Top1 | DCASE mAP | ENABird mAP |
|---|---|---|---|---|---|
| Baselines | | | | | |
| SVM | 13.90 | 77.90 | **91.40** | 14.60 | 29.90 |
| rn50p | 54.80 | 67.30 | 76.30 | 17.80 | 42.40 |
| rn152p | 57.30 | 66.20 | 74.10 | 19.80 | 42.90 |
| vggish | 44.00 | 80.80 | 90.60 | 33.50 | 53.50 |
| iNatSounds models | | | | | |
| MV3 | 66.91 | 81.87 | 87.05 | 34.65 | 62.26 |
| R50 | 68.45 | 74.88 | 85.61 | 33.46 | **70.09** |
| ViT | **70.41** | **84.40** | 89.21 | **42.37** | 60.06 |

Table A5: **Performance on BirdCLEF [46]**. We compare models trained from scratch, pretrained on ImageNet and pretrained on iNatSounds. We split BirdCLEF 2024 train set into 80:20 train:val.

| Model | PreTraining | Class-average Top1 | Simple Top1 |
|---|---|---|---|
| | scratch | 03.58 | 13.46 |
| MV3 | ImageNet | 09.08 | 31.52 |
| | iNatSounds | **12.18** | **40.33** |
| | scratch | 01.57 | 05.92 |
| R50 | ImageNet | 08.71 | 31.33 |
| | iNatSounds | **11.57** | **41.26** |
| | scratch | 00.88 | 03.31 |
| ViT | ImageNet | 01.96 | 07.36 |
| | iNatSounds | **11.95** | **38.70** |

