# OpenReview forum: "The iNaturalist Sounds Dataset"
_NeurIPS.cc/2024/Datasets_and_Benchmarks_Track — NeurIPS 2024 Track Datasets and Benchmarks Poster_

### Official Review · Reviewer_u4zG · 2024-07-24

**Rating:** 8
**Confidence:** 4
**Clarity:** Yes, the paper’s generally very clear.

**Review:**

Overall, I think this is a strong contribution to the community and commend the authors’ effort. I hope the notes below will be helpful in revising the paper.

There are a number of decisions made in the preparation of this dataset which require justification:
- “remove observations with audio having a sampling rate exceeding 48kHz” -> It’s trivial to downsample these to 48kHz, so it’s not clear why they would be removed. This seems to limit the dataset without cause?
- “resample the remaining audio files to 22.05 kHz” 22.05kHz is quite low, considering that some animals produce sound above its Nyquist frequency. For example, rodent vocalizations seem to extend up to around 20kHz, even if human audibility is the criterion (which it’s not clear that it should be, given the stated application areas like bioacoustic monitoring). I strongly recommend releasing it with the original sample rates, considering that it’s trivial to resample audio in-memory these days.
- “store them as single-channel WAV files” -> once again, reducing multichannel audio to monaural audio is trivial to do in-memory for downstream use. It would be simple to preserve the original channel counts, since the spatial dimension could be useful in some cases.

These seem to be decisions made in the interest of uniformity (with the exception of discarding >48kHz-sampled signals, which seems unnecessary even for this since they could be easily resampled as the 48kHz, 44.1kHz, etc. signals would be), but they are contrary to the goal of bioacoustic diversity since they limit this.

The temporal split is clever when considering seasonality, but introduces potential biases: platforms tend to accumulate more users over time (so the pool of recordists is likely different, which could come with geographical differences, etc.), and recording gear also proliferates over time. This is reflected in the unusually small differences in size between the splits. Typically, validation and test splits are much much smaller. For example, the train split here is ~59% (in examples; given the non-uniform lengths, hours would be preferable) vs. about 99% (unbalanced) for AudioSet or ~92% for VGGSound, etc. Additionally, the pandemic might have shifted the distribution of recordings to some degree, which a temporal split would be vulnerable to. Could some analysis be done to evaluate the representativeness of the validation and test distributions?

I strongly recommend removing the claim of “a bioacoustic foundation model.” It seems like what is actually checked is whether it is a good pretraining dataset, which is conceptually very different. A foundation model is likely to have capabilities far beyond classification, unless this term is being used in a non-standard way.

**Strengths:**

As I mentioned, I think this is a strong contribution. The scale of the dataset, and the demonstration of its value in pretraining, is very helpful. The implemented experiments are well-designed and clearly establish the value of the dataset, and the paper is pretty well-written too.

**Additional Feedback:**

N/A

**Correctness:**

Yes, in general I think the claims are appropriate, the dataset is well constructed, and the evaluation is well designed and implemented.

**Documentation:**

Yes.

**Ethics:**

I don’t think this paper introduces new, substantive ethical concerns. To the degree it’s important for the paper to discuss this, I think it’s done so (e.g. geographical bias, etc. discussed in section 6).

**Limitations:**

Yes, I think the limitations are clear in the paper, and the authors have taken a modest and well-scoped approach to the claims overall.

**Opportunities For Improvement:**

As noted in my review, I think a few things can be clarified. If a version of the dataset without the preprocessing can be released, I think that would be very useful as well.

**Relation To Prior Work:**

Yes, right from the beginning (Figure 1 gives context immediately!).

**Summary And Contributions:**

This paper contributes a large and diverse dataset of weakly labeled animal sounds sourced from a citizen science platform, and conducts a series of classification-oriented experiments to show how this can serve as a useful pretraining dataset and a task in itself.

---

> ### Author Rebuttal · Authors · 2024-08-17
>
> **[u4zG] Filtering based on Sampling rate.**
>
> Some species of bats and whales in the dataset had sounds that were naturally beyond 48kHz. Sub-sampling recordings of these species would likely lead to loss of the primary signal. Thus, we instead choose to remove these recordings altogether. This filtering led to a drop of around 1.2\% of total observations, which seemed reasonable.
>
> **[u4zG] Re-sampling to 22.05 kHz and single-channel WAV. Un-processed data.**
>
> We made these choices for uniformity to make it easier for ML researchers to train and evaluate models. For example, standardizing audio as (spectrogram) images simplifies the process for CV researchers to work on problems in the audio domain. Our final dataset release will include links to the original data and Python scripts to generate different variants, allowing end users to experiment with these design decisions.
>
>
> **[u4zG] Representativeness of Training/Val/Test Splits.**
>
> As seen in Figure 2 of the paper, the validation and test splits consist of the more frequent species in the training set. The validation and test splits are similar and have roughly the same distribution of taxonomic classes. We also computed the Pearson correlation coefficient between the geographic distributions (per-class mean latitude and longitude) and found it to be high among the common species in the training and validation or test sets.
>
>     Train and Val 	Latitude: 99.41\% Longitude: 99.50\%
>     Train and Test 	Latitude: 99.41\% Longitude: 99.31\%
>
> Overall, we believe that the validation and test data are good representations of the iNaturalist training set. However, we agree that this is just a small portion of the vast world of natural sounds.
>
> **[u4zG] Bias and Size of Splits.**
>
> We agree with the reviewer about the biases introduced by a temporal split. However, we believe that the benefits of making the dataset extensible in a non-overlapping manner are highly valuable, as this allows for meaningful comparisons between models trained on different versions of the dataset. This mirrors the construction of the popular iNaturalist image classification datasets. Additionally, random sampling can introduce its own biases.
>
> We also agree that our validation and test splits are large compared to datasets of comparable scale. However, we found that a large validation split is necessary to reliably compare models and select hyperparameters. In competitions, once the hyperparameters have been narrowed down, it is common practice to combine the validation set with the training set to achieve the best test performance. With the growth of observations on the platform, this will become less of an issue in future years.
>
> **[u4zG] Foundation model claim.**
>
> We agree that the primary contribution is a foundation dataset, but, to our dismay, the literature tends to use the terms foundation model and foundation dataset somewhat interchangeably. We will rephrase this claim in the revised version.

---

> > ### Comment · Reviewer_u4zG · 2024-08-17
> >
> > Thank you for these helpful responses. A few specific follow-ups:
> > - On filtering: Note that the Nyquist frequency of a 22.05kHz sampling rate is ~11kHz. This means sounds above this would be decimated. Filtering content above 48kHz still leaves over two octaves of sound filtered out here. Are any sounds or categories affected by this? Additionally, discarding sounds sampled >48kHz establishes a Nyquist frequency of 24kHz, for the first part, not 48kHz.
> > - On filtering and preprocessing: Keeping the original data available and making the preprocessing reproducible is very helpful. Will the original data be pre-filtering? I imagine the bat and whale sounds would still be useful to many.
> > - On sampling: "Additionally, random sampling can introduce its own biases" -> In what way? Do you mean introduce biases by ignoring the seasonality present in the data?

---

> > > ### Comment · Reviewer_u4zG · 2024-08-17
> > >
> > > Also, one missing reference I should have mentioned earlier but missed in my review is biolingual:
> > > > Robinson, David, Adelaide Robinson, and Lily Akrapongpisak. "Transferable models for bioacoustics with human language supervision." ICASSP 2024-2024 IEEE International Conference on Acoustics, Speech and Signal Processing (ICASSP). IEEE, 2024.
> > >
> > > This introduces a dataset called AnimalSpeak, which includes sounds from iNaturalist and other platforms (Xeno-canto, etc.). A brief comparison would be very helpful to the reader.

---

> > ### Author Rebuttal · Authors · 2024-08-27
> >
> > **Filtering and preprocessing.**
> >
> > We appreciate the reviewer’s insightful observations regarding our filtering choices. Our primary objectives with the iNatSounds dataset are twofold: (1) to make it highly accessible to the broader machine learning community, and (2) to align it with the near-term goals of iNaturalist.
> >
> > Regarding accessibility, we standardized the dataset by converting all audio files to the WAV format. This choice eliminates the need for codec support and ensures compatibility with tools like SciPy, TensorFlow, and PyTorch, which have native support for WAV files. While we acknowledge that WAV files are uncompressed and thus larger, we mitigated this by resampling the audio to mono at a 22.05kHz sampling rate. This approach reduces the dataset size and minimizes potential issues stemming from variations in audio parameters. We recognize that this results in the loss of higher frequency signals, but we believe that the consistency and ease of use provided by this standardization outweigh these costs. Researchers focused on specific taxa requiring the original signal can access the user-contributed files via the iNaturalist platform.
> >
> > For goal (2), focusing on taxa with audio recorded at 48kHz or lower aligns with iNaturalist’s data collection, which primarily relies on mobile phones—most of which do not support recording above 48kHz. We excluded files with higher sample rates because taxa requiring such rates are unlikely to be effectively recorded by typical mobile devices, thus making their inclusion in this dataset less practical.
> >
> > **Biases in random sampling.**
> >
> > Observations from iNaturalist inherently exhibit geospatial and temporal biases, particularly towards population centers in the United States and the northern hemisphere spring. A random sampling of iNaturalist data would naturally reflect these biases. While we do not claim that our sampling strategy fully eliminates them, we have designed it to preferentially select observations that are more globally and temporally distributed, to the extent possible.
> >
> > **Biolingual reference and AnimalSpeak.**
> >
> > Thank you for the reference, we will add it to the revised version and include a comparison with AnimalSpeak in our related work.

---

> > > ### Comment · Reviewer_u4zG · 2024-08-27
> > >
> > > Thank you for addressing my comments. I still need to strongly disagree on the filtering/resampling strategy. Prematurely resampling has negligible usability benefit, in my opinion, but a non-negligible quality cost. I think it’s worth considering a release strategy that preserves this information, if possible, or provides clear and replicable instructions on obtaining the originals. The point about mobile devices is taken, but the 22.05kHz sampling rate filters out much content that is captured by such devices as well. I appreciate your responses on the other points and have no further comments there.

---

### Official Review · Reviewer_XHtJ · 2024-07-25
**A New Multi-Taxa Dataset**

**Rating:** 8
**Confidence:** 5
**Clarity:** The paper is clearly written.

**Review:**

So, I am very excited about this dataset, but have some issues with the write-up.

In the related works, the relationship to previous work on bioacoustics is lacking.

First, Xeno-Canto is the most comparable dataset to iNatSounds, but is omitted. A compare/contrast with that dataset is required, as it is the closest point of comparison and a commonly used training set for bioacoustic models.

Second, acoustic datasets are generally classified as fully-annotated (expert listeners annotate all target sounds in the recordings), weakly annotated (usually some kind of positive-only labeling, for one or more species, and/or non-timestamped labels), or unlabeled. The scale of these datasets varies with the amount of annotation effort put in: fully annotated datasets are often only dozens of hours, while unannotated datasets may contain millions of hours of audio (eg, the Australian Acoustic Observatory). The references for 'other existing benchmarks for birds' are relatively small, but they are strongly annotated and provide strong evaluation signals for a diversity of world regions. These are serving different goals than large weakly labeled datasets like iNatSounds.

Another consideration in citizen science datasets is sample bias, which can manifest in a number of ways, such as sampling near human population centers, over-representation of alarm calls, and omission of harder to distinguish or less common calls. Fully annotated datasets help us understand model performance in passively recorded audio, which is somewhat less subject to these limitations.

As a result, I tend to trust results of evaluation on fully-annotated datasets more than held-out sets from citizen-science datasets, though, of course, it's still fine to report performance on held-out sets.

We also recommend using class-averaged ROC-AUC as an evaluation metric. Average precision is biased by label prevalence, which hinders comparison of scores across different species.

Additionally, metrics which require choice of a detection threshold (like F1); the thresholding strategy should be detailed in your model section. It seems like you've chosen the threshold which gives the best F1 score on the validation set; this provides an upper-bound on performance, but may not be informative about model performance in practice.

"ignore species outputs that do not overlap with the 1212 val/test species when doing evaluation" - This is a bit strange; users of the model will typically not know which species appear in their datasets... If the model gives a high score for a non-test species, this misclassification is disregarded, leading to inflated model scores. (Restriction to loosely-defined geographically feasible species sets is more defensible, if you want to avoid the full all-against-all problem. But again, this is something that evaluation on fully-annotated datasets can help with.)

**Strengths:**

The dataset is a large and welcome addition to the bioacoustics space. As discussed below, it looks like an excellent dataset to expand on existing sources for supervised training. The diversity of taxa represented in the dataset is particularly salient, and differentiates it from other sources.

**Additional Feedback:**

N/A

**Correctness:**

The dataset is constructed in a sound way.

I am not sure I would recommend this as a benchmark, due to the weakly labeled nature of the data. A benchmark consisting of training on iNatSounds and testing on some existing eval datasets (eg, BEANS + fully annotated bird datasets) would be reasonable.

**Documentation:**

The github page exists and provides clear links to the data. Thanks!

**Ethics:**

No issues.

**Limitations:**

The paper includes a limitations section, which addresses the main issues.

**Opportunities For Improvement:**

Adding a comparison to Xeno-Canto and a more structured view of other bioacoustic datasets is important for properly situating this work in the literature.

The biggest limitation of the dataset is the weak labelling and sample bias, both of which are acknowledged. To my mind, this makes the dataset valuable as an additional training data source, but limits its utility as a validation dataset. So, it would be valuable to understand the performance of models trained on iNatSounds on more established eval sets, such as the various BirdCLEF datasets and the BEANS benchmark. Both SWAMP and SSW60 are Sapsucker Woods datasets, and SSW60 is weakly labeled - I would drop that one. It would be great to include some data from other world regions in the eval, given the heavy North American sample bias present in the training data (eg, the recent BirdSet (https://arxiv.org/pdf/2403.10380) pulls together past BirdCLEF test data and includes a good amount of South American data).

Some additional summary information on the dataset would be helpful; eg, total length of audio in hours, and some stats for understanding the distribution of audio file lengths (mean, median, max, at least, but a histogram would be nice).

For allaying the concerns about evaluation on weakly labeled data, you could randomly sample some data to estimate the prevalence of unlabeled sounds.

**Relation To Prior Work:**

This is discussed above in the Review section.

**Summary And Contributions:**

The iNatSounds database provides a large number of acoustic observations for over 5k species, covering a broad range of taxa. A yearly train/val/test split is presented, and baseline model scores are provided. The trained models are also evaluated on a couple additional datasets.

---

> ### Author Rebuttal · Authors · 2024-08-17
>
> **[XHtJ] Comparison to Xeno-Canto.**
>
> Thank you for pointing this out, and we apologize for not referencing Xeno-canto alongside iNaturalist and the Macaulay Library. However, it’s important to note that while Xeno-canto is a valuable data source, it’s not a machine-learning-ready dataset just like iNaturalist or Macaulay library. One of our key contributions is the creation of a dataset and benchmark through meticulous data curation, which we have made publicly available with appropriate licensing terms. Thus our comparisons are limited to similar datasets. While some models, like BirdNet, do utilize data from Xeno-canto and the Macaulay Library for training, they have not publicly released their training datasets, likely due to licensing constraints. However, as the reviewer noted there are concurrent efforts, such as BirdSet which uses Xeno-canto but focuses on around 400 species of birds and has a different geographical distribution than iNaturalist. We will include a comparison to Xeno-canto and a structured overview of bioacoustic datasets, focusing on various aspects such as strong vs. weak labels, and taxonomic and geographic diversity, in the revised version of the paper.
>
> **[XHtJ] Discussion on other large, weakly labeled datasets.**
>
> We would appreciate references for datasets or surveys that we could add to the comparison figure or as references. Thank you.
>
> **[XHtJ] Sample bias: citizen science vs passively recorded datasets.**
>
> We agree with the reviewer and have designed the temporal splitting and geo-temporal subsampling to reduce sample bias. While passively recorded datasets generally have lower bias, they also tend to be less diverse in terms of taxa and geographical distribution. However, future work could explore ways to benefit from both types of data, and one of the goals of our work is to maintain and extend this benchmark regularly by incorporating observations from iNaturalist and elsewhere based on community feedback.
>
> **[XHtJ] ROC-AUC as an evaluation metric.**
>
> Our ROC-AUC numbers are ~99% for most models and we did not include them as it was hard to discern differences between the models using that metric. For completeness we have included them below and will add them to the Appendix :
>
> | Model | val | test | val + geo | test + geo |
> | :- | :-: | :-: | :-: | :-: |
> | MV3 | 98.7 | 98.6 | 99.2 | 99.2 |
> | R50 | 98.9 | 98.8 | 99.3 | 99.2 |
> | ViT | 99.0 | 98.9 | 99.4 | 99.4 |
> | | | | | |
>
> **[XHtJ] Thresholding strategy.**
>
> Yes, the best F1 score is obtained by selecting the optimal threshold on the evaluation set. This is a commonly used metric [1, 2] . While this approach simplifies evaluation, we agree that the real-world deployment of these models requires additional model calibration or careful threshold selection.
>
> [1] Cañas, Juan Sebastián, et al. "A dataset for benchmarking Neotropical anuran calls identification in passive acoustic monitoring." Scientific Data 2023
>
> [2] Kahl, Stefan, et al. "BirdNET: A deep learning solution for avian diversity monitoring." Ecological Informatics 2021
>
> **[XHtJ] Ignoring non-overlapping species outputs.**
>
> This simulates the closed-set assumption in most benchmarks where the set of target classes are known ahead of time. For completeness, we have also included results without this masking below. We have also added the results of our model without masking but including Geo-Prior where the performance is better than the model with masking (but without the Geo Prior):
>
> | Model | With | Masking | Without | Masking |Without Masking  | + GeoPrior |
> | - | :-: | :-: | :-: | :-: | :-: | :-: |
> | |**Top 1**|**Top 5**|**Top 1**|**Top 5**|**Top 1**|**Top 5**|
> | MV3 | 50.2 | 70.5 | 43.8 | 63.7 | 53.2 | 73.2 |
> | R50 | 53.4 | 74.6 | 44.7 | 65.9 | 55.6 | 75.1 |
> | ViT | 55.1 | 74.9 | 53.5 | 74.1 | 61.6 | 80.5 |
> | | | | | | | |
>
>
> **[XHtJ] Performance on BirdCLEF and BEANS.**
>
> Thank you for the suggestions of BirdCLEF and BEANS. We are happy to report the iNatSounds pre-trained models compare favorably to ImageNet pre-trained ones as well as baselines reported in the paper.
>
> Performance on 5 datasets part of BEANS. These are models pre-trained with iNatSounds and fine-tuned on a dataset train-set.
>
>
> | Model | CBI (ACC) | HumBugDB (ACC) | Dogs (ACC) | DCASE (mAP) | ENABird (mAP) |
> | :- | :-: | :-: | :-: | :-: | :-: |
> | | | **Baselines** | | | |
> | SVM | 13.90 | 77.90 | 91.40 | 14.60 | 29.90 |
> | rn50p | 54.80 | 67.30 | 76.30 | 17.80 | 42.40 |
> | rn152p | 57.30 | 66.20 | 74.10 | 19.80 | 42.90 |
> | vggish | 44.00 | 80.80 | 90.60 | 33.50 | 53.50 |
> | | | **iNatSounds** | | | |
> | MV3 | 66.91 | 81.87 | 87.05 | 34.65 | 62.26 |
> | R50 | 68.45 | 74.88 | 85.61 | 33.46 | 70.09 |
> | ViT | 70.41 | 84.40 | 89.21 | 42.37 | 60.06 |
> | | | | | | |
>
> Performance on BirdCLEF: We compare models trained from scratch, pretrained on ImageNet and pretrained on iNatSounds. We split BirdCLEF 2024 train set into 80:20 train:val (the test labels are not publicly available).
>
> | Model | Training | Class-average ACC | Simple ACC |
> | :- | :-: | :-: | :-: |
> | | scratch | 03.58 | 13.46 |
> | MV3 | ImageNet | 09.08 | 31.52 |
> | | iNatSounds | 12.18 | 40.33 |
> | | | | |
> | | scratch | 01.57 | 05.92 |
> | R50 | ImageNet | 08.71 | 31.33 |
> | | iNatSounds | 11.57 | 41.26 |
> | | | | |
> | | scratch | 00.88 | 03.31 |
> | ViT | ImageNet | 01.96 | 07.36 |
> | | iNatSounds | 11.95 | 38.70 |
> | | | | |
>
> **[XHtJ] Dataset summary information.**
>
> We will include the total length of audio in iNatSounds in the introduction (shown in Fig1). We will include statistics of audio durations (mean: 19.52s, median: 13.94s, max: 33m 34.82s and (PFA) histogram) in the revision, and are also shown in the attached PDF.
>
> **[XHtJ] Prevalence of unlabeled sounds.**
>
> Great suggestion. We did some preliminary analysis of the type seen in Figure 4 of the paper where we did observe species other than the primary one being present. But we haven’t done a systematic analysis. We will aim to add this in the revised version.

---

> > ### Comment · Reviewer_XHtJ · 2024-08-18
> >
> > Thanks to the authors for taking my comments under consideration; I'll be gladly increasing my recommendation based on the stated revisions.
> >
> > "Our ROC-AUC numbers are ~99% for most models..."
> > "We split BirdCLEF 2024 train set into 80:20 train:val (the test labels are not publicly available)."
> >
> > These statements are indicative of the main point I want to get across: Large weakly-labeled human datasets have fundamentally different distributions than 'real world' data. Hard examples often go unidentified or don't get uploaded, so these sets tend to consist of easy examples, and, as a result, ML-standard train/test split strategies tend to systematically overestimate model quality. BirdCLEF 2024 is an excellent example: Participants also saw very high performance on a cross-validation split of the training data, yet the high score on the test set was only about 0.69 ROC-AUC. If you would like a more realistic scenario, the test data from past BirdCLEF competitions is freely available on Zenodo, and has recently been collected in the BirdSet dataset.
> >
> > Now, this is arguably a much bigger question than you're trying to answer with this dataset, and I don't think it /can/ be addressed with any single dataset. But I think it should inform the narrative. The train/test split tells us how we can expect models to perform on future research-grade iNat samples, but may not be helpful for understanding model performance outside of that narrow context.
> >
> > Inclusion of benchmark model performance on other benchmarks (like BEANS and/or past BirdCLEF /test/ sets) is a good way of getting at that question, and I thank the authors for running the additional evaluations.

---

> > > ### Author Rebuttal · Authors · 2024-08-27
> > >
> > > Thank you for the note. We agree that the proposed benchmark only goes so far in predicting performance in real-world deployment, where there might be a considerable domain shift. We will look into the BirdCLEF and BirdSet datasets for further evaluation in the future.

---

### Official Review · Reviewer_geSw · 2024-08-04
**Large sound collection**

**Rating:** 6
**Confidence:** 4
**Clarity:** The article is well structured and cl…

**Review:**

The work demonstrates high quality in terms of creating a larger dataset with sounds of over 5500 species and a diversity of species included. The benchmarking of different backbone architectures, training, and presenting results showcase a rigorous evaluation process.
The article is well structured, providing clear information about the dataset, the sources of the dataset, and methods to evaluate the dataset. Results are compared and discussed with clarity and figures are also included where necessary.
The compilation of dataset fills a significant gap in the availability of large scale fine grained soundscape acoustic dataset. The comparison with other related datasets highlights the uniqueness and originality of the work.
The work is highly significant because it aids in advancing research in bioacoustics, diversity monitoring, and applications of sound recognition in artificial intelligence.
The con of the dataset is weak labelling for some of the instances. It would be useful to provide strong labelling for a subset of the dataset.

**Strengths:**

- A large-scale dataset with over 230000 audios of 5500 different species.
- Global contributions from 27000 recordists, enhancing dataset diversity.
- Benchmarking of backbone architectures and pretraining model for downstream tasks.
- Potential of applications in biodiversity monitoring and conservation.

**Additional Feedback:**

Please see above.

**Correctness:**

This article presents a dataset that benchmarks the backbone models and compares the results with related models and datasets.

**Documentation:**

Sufficient details are provided on data collection and organization. Data is freely available to promote research in the domain of bioacoustics. It also includes the licensing information indicating the terms under which the data can be used and shared. The following URL is given to access the dataset: https://www.inaturalist.org

**Limitations:**

Yes, the authors have addressed the limitations of the work and suggested future work as well.

**Opportunities For Improvement:**

The paper provides a rich dataset with a large number of records and reasonable performance. Nevertheless, the quality and robustness of the dataset have not been discussed extensively. It is important to highlight the inter-agreement between naturalists, enabling quality control. Furthermore, it will be good to provide a subset of strong labelling for broader impacts.

**Relation To Prior Work:**

This article presents the iNaturalist Sounds Dataset, which is different from previous work in several key aspects, including Dataset size, Diversity, Global Contribution, Benchmarking, Evaluation, and addressing the gap in acoustics datasets.

**Summary And Contributions:**

The “iNatSounds” dataset is a comprehensive collection of more than 230000 audio files containing audio sounds of more than 5500 species including birds, mammals, insects, reptiles, and amphibians. These sounds are collected worldwide and are labelled with species names generating that specific sound. This article benchmarks various backbone architectures and compares the multi-label and multi-class results. The achieved results demonstrated the dataset utility for pretraining and achieving high performance on downstream evaluation task. Benchmarking different backbone models and presenting a larger open-source dataset are the main contributions.

---

> ### Author Rebuttal · Authors · 2024-08-17
>
> **[geSw] Quality and robustness of dataset / Inter-agreement between naturalists.**
>
> Recordings in iNatSounds are taken from “research-grade” observations in iNaturalist, meaning the community has reached a consensus (at least ⅔ agreement) on the species ID or lower  and the observation passes certain quality control tests. Observations can also lose research grade status if an inaccuracy is flagged by a naturalist and it will be removed from future iterations of iNatSounds. These are described in https://www.inaturalist.org/pages/archived+help#quality and we will add a discussion about the data quality control measures in the revision.
>
> **[geSw] Subset of strong labeling for broader impacts.**
>
> We agree that the weakly labeled data is a limitation of iNatSounds, but labeling a subset with strong labels might be feasible only in the future. This would require better labeling tools and engaging with the expert community on iNaturalist and elsewhere. However, we are comforted by the fact that strong performance on iNatSounds is a good indicator of downstream performance on a range of datasets. We have added some additional results on the BEANS benchmark in the rebuttal where our models also outperform prior work and baselines.

---

> ### Author Response · Authors · 2024-08-27
>
> As the discussion period is nearing its end, we kindly ask if the reviewer could review our rebuttal and share any additional concerns.

---

### Author Rebuttal · Authors · 2024-08-17

We thank the reviewers for their constructive feedback. We are happy to hear that they are excited (XHtJ) about this work. We thank the reviewers for appreciating the scale and diversity (geSw, XHtJ, u4zG) of data in iNatSounds; recognizing its applications and impact (geSw, u4zG); and commending on our experimental design (geSw, u4zG) and writing style (geSw, XHtJ, u4zG). We appreciate the reviewers’ suggestions to alleviate some limitations in the dataset and better place this work in relation to prior art. We address specific concerns raised by reviewers as separate responses.

---

### Decision · Program_Chairs · 2024-09-26

**Decision:**

Accept (Poster)

**Comment:**

This paper presents iNatSounds, a collection of audio recordings from iNaturalist that covers over 5,000 species and 230,000 audio recordings. The data is split into training, validation and test splits according to years in which the data was released. A variety of models are evaluated on this data using both multi-class and multi-label training. These models show strong performance in downstream tasks.

The reviewers appreciate the scale and coverage of the dataset, the writing, and the well-designed experiments that cover multiple models.

But the reviewers were not without concerns. Some worry about the resampling of the audio to 22.05 kHz, the weak labels, and the potential sample bias in citizen science projects. It is also important to note that this data doesn't necessarily measure performance of models in many "real-world" contexts (e.g., passive acoustic monitoring projects) because the nature of the data is so different compared to iNaturalist research-grade observations. This weakens the argument that models trained on this dataset can be considered "bioacoustics foundation models".

However, overall this dataset is thoughtfully constructed and the paper was well received by reviewers. I recommend to accept it.